# Direct Comparison of HPV16 Viral Genomic Integration, Copy Loss, and Structural Variants in Oropharyngeal and Uterine Cervical Cancers Reveal Distinct Relationships to E2 Disruption and Somatic Alteration

**DOI:** 10.3390/cancers14184488

**Published:** 2022-09-16

**Authors:** Travis P. Schrank, Sulgi Kim, Hina Rehmani, Aditi Kothari, Di Wu, Wendell G. Yarbrough, Natalia Issaeva

**Affiliations:** 1Department of Otolaryngology/Head and Neck Surgery, School of Medicine, The University of North Carolina, 170 Manning Drive, Chapel Hill, NC 27599, USA; 2Lineberger Cancer Center, UNC, Chapel Hill, NC 27599, USA; 3Department of Biostatistics, The University of North Carolina at Chapel Hill, Chapel Hill, NC 27599, USA; 4Division of Oral and Craniofacial Health Sciences, Adams School of Dentistry, School of Medicine, The University of North Carolina at Chapel Hill, Chapel Hill, NC 27599, USA; 5Department of Pathology and Lab Medicine, UNC, Chapel Hill, NC 27599, USA

**Keywords:** HPV16, oropharynx, squamous cell carcinoma, uterine cervix, integration

## Abstract

**Simple Summary:**

HPV16 causes approximately 60% of uterine cervical cancer and 95% of HPV-driven oropharynx cancers. Despite both being HPV-associated, these tumors are very different. We directly compare integration of the HPV16 genome into the human genome in these two diseases, finding that the viral gene E2 is frequently lost in cervical cancer, but usually maintained in oropharyngeal cancer. We also found that oropharyngeal cancers with integration have many more integration sites per tumor and these more frequently occur in genomic regions with a high density of genes.

**Abstract:**

Squamous cell carcinoma of the oropharynx caused by HPV type 16 (HPV16+ OPSCC) is the most common HPV-associated malignancy in the USA and has many molecular differences from uterine cervical squamous cell carcinoma (UCSCC). Our understanding of HPV oncogenesis relied on studies of UCSCC revealing a consensus model reliant on HPV integration with a loss of E2. Here, we compare patterns of HPV integration in UCSCC and OPSCC by analysis of affinity capture sequencing of the HPV16 genome in 104 OPSCC and 44 UCSCC tumors. These cohorts were contemporaneously sequenced using an identical strategy. Integration was identified using discordant read pair clustering and assembly-based approaches. Viral integration sites, structural variants, and copy losses were examined. While large-scale deep losses of HPV16 genes were common in UCSCC and were associated with E2 loss, deep copy losses of the HPV16 genome were infrequent in HPV16+ OPSCC. Similarly, structural variants within HPV16 favored E2 loss in UCSCC but not OPSCC. HPV16 integration sites were non-random, with recurrent integration hot-spots identified. OPSCC tumors had many more integration sites per tumor when compared to UCSCC and had more integration sites in genomic regions with high gene density. These data show that viral integration and E2 disruption are distinct in UCSCC and OPSCC. Our findings also add to growing literature suggesting that HPV tumorigenesis in OPSCC does not follow the model developed based on UCSCC.

## 1. Introduction

Papillomaviruses are species-specific, non-enveloped, double-stranded DNA viruses with an about 8kb circular genome protected by a 55 nm icosahedral capsid [1,2,3,4]. Human papillomaviruses (HPV) are associated with benign and malignant epithelial lesions. Subsets of HPV are high-risk and are causative agents of cervical cancer [5,6] and the majority of oropharyngeal squamous cell carcinomas (OPSCC) [7,8,9,10]. Both cervical and oropharyngeal cancers share universal risk factors that involve sexual behavior and cigarette smoking.

Cervical cancer, the second most common cancer in woman and with a gradually decreasing incident rate in the United States, is usually associated with HPV. Development of cervical cancer is a multistep process that includes initial HPV infection followed by persistent infection resulting in a relatively well-established progression from pre-cancerous cervical intraepithelial neoplasia (CIN) to malignant lesions [11]. In early lesions, the HPV genome remains in an episomal state, but with advancement to higher grade lesions, HPV DNA frequently integrates into the host genome, facilitating cancer development [12,13]. Integration accelerates carcinogenesis through frequent disruption of the HPV E2 gene, causing increased expression of the major oncogenes, E6 and E7, that directly inhibit the key human tumor suppressors, p53 and Rb [14,15]. Maintenance of HPV oncogenes is critical, but other HPV genes are variably lost [16].

As the opposite of cervical cancer because it is constantly increasing in the USA, oropharyngeal cancer is detected mostly in men, with male-to-female ratio of about 4:1, and roughly 80% of this disease is driven by HPV. HPV tumorigenesis in the oropharynx has not been as intensively investigated, and it most likely differs from cervical lesions due to unique features of tonsillar epithelia. The epithelia of the lingual and palatine tonsils, the regions where HPV-associated cancer arises [17], are lymphoid-associated epithelia with specialized crypts that facilitate antigen presentation. Tonsils are well recognized sites for pathogenic viral infections, including Epstein–Barr virus [18], adenoviruses, influenza and parainfluenza viruses, herpes simplex virus, and respiratory syncytial virus [19]. Natural susceptibility to a viral infection of tonsils most likely explains why HPV-associated head and neck cancer nearly always arises in this anatomical site. In contrast to cervical cancer, defining particular steps of HPV-carcinogenesis in tonsillar epithelia has been hampered by the absence of clinically detectable pre-malignant lesions and established screening procedures [11].

In addition to epidemiological differences, although initiated by the infection of HPV, cervical and oropharyngeal malignancies are characterized by several biological and clinical dissimilarities that highlight distinct HPV-mediated carcinogenesis [20]. First, HPV16 and HPV18 account for approximately 75% of uterine cervical tumors, but HPV type 16 is responsible for roughly 92% of OPSCC, and HPV type 18 is extremely rare in OPSCC [21]. Second, HPV-associated OPSCC has distinct cellular gene and protein expression profiles compared to cervical cancer [22,23]. Third, patients with HPV-positive OPSCC generally respond better to treatment than patients with HPV-positive cervical cancer [11]. Fourth, HPV-positive OPSCCs harbor a different somatic mutation spectrum than cervical cancer [24,25]. Lastly, fifth, ~40% of oropharyngeal cancers display constitutively active NF-κB, mostly due to genetic inactivation of *TRAF3* and *CYLD* genes that are rare in uterine cervical cancer [26]. The NF-kB active subtype of OPSCC lacks HPV integration and has improved survival compared to OPSCC with low NF-kB activity [27,28,29,30]. Moreover, a growing amount of data point out a remarkable diversity of tumor microenvironment with increased CD4+:CD8+ T cell ratio in oropharyngeal as compared to cervical cancer [31].

Both integrated and episomal HPV DNA occur in OPSCC and cervical malignancies, with some tumors containing only integrated HPV DNA, others containing only episomal DNA, and a third group containing both [25,32,33,34,35]. Since integration is implicated in HPV carcinogenesis and is found in more than 50% of both cervical and oropharyngeal cancers, recent studies have focused on mapping integration sites and exploring how integration contributes to carcinogenesis. Key findings from these studies include a wide number of mapped integration sites, recurrent integration hotspots, and discoveries of accompanying complex structural genomic rearrangements associated with modified expression of affected genes [25,27]. In addition, a recent study found non-chromosomal circular viral/human DNA with oncogenic properties in OPSCC [36].

Important differences between HPV driven oropharyngeal and uterine cervical carcinomas continue to emerge; however, no comprehensive study has compared HPV integration sites and HPV structural genomic aberrations present in these two cancers. To define similarities and differences, we used identical ultra-deep targeted sequencing of HPV16 from 104 oropharyngeal and 44 uterine cervical tumors. Here, we present an in-depth investigation that reveals drastic dissimilarities in HPV integrations and structural genomic aberrations in OPSCC and UCSCC. These findings emphasize their individual biology and suggest that HPV-driven carcinogenesis is different between these tumor types.

## 2. Materials and Methods

***Human Subjects***—DNA sequencing of data was performed as a part of the UNCseq tumor sequencing program. Tumor sample identifiers and genomic data were derived from the clinical trial LCCC1108: *Development of a Tumor Molecular Analyses Program and Its Use to Support Treatment Decisions.* This IRB-approved trial opened in 2011. All studies were done with the approval of our Institutional Review Board, patient participation required written informed consent, and all studies were conducted in accordance with recognized ethical guidelines as described in U.S Common Rule. By means of a chart review of electronic medical records, demographic information was obtained for each study subject, including age, gender, race, and smoking history. The clinical stage at presentation according to the AJCC staging system (AJCC 8th edition) was recorded considering that many patients did not receive pathological staging.

***DNA Isolation, Library Preparation, and Sequencing***—A pathologist examined H&E-stained slides from each case to confirm the diagnosis of squamous cell carcinoma. Automated DNA extraction was from FFPE tissue sections using the Promega Maxwell MDx16^TM^ instruments (Promega) and then fragmented by sonication. Subsequent quality assessments were performed by ultraviolet absorbance and quantity assessments. During DNA isolation and library preparation, DNA concentration was measured by fluorometry and DNA quality was evaluated using the Agilent 2100 Bioanalyzer high sensitivity assay. DNA libraries were pooled for deep sequencing using an Illumina HiSeq2500^TM^ sequencer. The UNCseq targeted sequencing platform involves sequencing exons of a custom list of 650 human genes (covering 3.4 M bases) and 10 pathogen genome segments in fixed or frozen cancer tissue and matched germline DNA from consenting local patients. This custom sequencing platform provided deep coverage of all HPV16 open reading frames. For general metrics on human gene sequencing quality, read depth, and coverage statistics, please see our recent report which reviewed these features of the data set [37]. Excluding the hypervariable region 3150–3351, the median coverage of the HPV16 genome was 7904 with an IQR of 6867–7953. The hypervariable region 3150–3351 had a median coverage of 3307 with an IQR of 837–7377.

***Inclusion Criteria***—The UNCseq database was queried for p16+ tumors originating from the anatomic oropharynx (tonsil or tongue base) with available tumor sequencing data as well as data on stage, treatment strategy, clinical outcome, and histopathology available. HPV16 positivity was confirmed by DNA sequencing reads which mapped to the HPV16 genome (see above comments). Patients were excluded from clinical analyses if tumors were not p16+. The UNCseq database was also queried for squamous cell carcinomas of the uterine cervix; HPV16 positivity was confirmed by DNA sequencing reads which mapped to the HPV16 genome (see Viral Copy Number Analysis section below). For both tumors from the oropharynx and uterine cervix, tumors with non-HPV16 genotypes were excluded from the study.

***Assigning HPV16 Positivity and Viral Copy Number Analysis***—Raw reads were aligned to the human genome plus a comprehensive library of HPV virus sequences, using compiled reference sequences from the ViFi analysis pipeline [38]. An HPV16 viral copy number was estimated based on the ratio of reads mapping to the HPV16 (NC_001526.4) genome and HG19. Log_10_(Reads_HPV16_/Reads_HG19_) > −4 was used as an empiric cut off for HPV positivity. This threshold was selected based on an obvious bimodal distribution in the data; see our prior publication justifying this criterion [21]. The Samtools Idxstats() function was used to quantify reads mapping to HPV16 and human chromosomes [39]. For HPV16-positive tumors, Log_10_(Reads_HPV16_/Reads_HG19_) calculated as above was considered as a relative metric of the HPV16 copy number.

***Integration Site Calling***—Raw reads were aligned to the human genome plus a comprehensive library of HPV virus sequences, using compiled reference sequences from the ViFi analysis pipeline [38]. The ViFi pipeline was used to identified discordant (human–viral) read pairs, which cluster to potential integration sites in the human and HPV genomes [38]. Breakpoints with more than 50 clustered discordant read pairs were classified as positive for integration. As a second method of detecting integration, an independent assembly structural variant caller, svABA, was utilized with default settings [40]. Alignment BAM files output from ViFi, which included hg19 as well as the HPV16 genome, were used as inputs for svABA. Contiguous sequences assembled by svABA were matched to discordant read pair clusters from ViFi. Discordant read pair clusters that were with fewer than 50 read pairs were also considered suggestive of integration if a continuous sequence supporting a HPV–human structural variant was also identified by svABA and passed the quality filter independently, or contained at least 5 reads spanning a HPV–human junction also identified by a ViFi cluster. Integration sites were identified from individual HPV–human breakpoints, looking for clusters of breakpoints within 1 Mb of each other using in-house scripts.

***Assigning Gene Effects***—The R SMITE [41] and GenomicRanges [42] packages were used to identify the nearest gene and minimum distance to it, for each high-confidence breakpoint. Based on prior reports demonstrating altered gene expression of genes as much as 500 Kb away from the gene of interest [7], we used ±500 kb as a threshold for genes whose functions may be affected by integration.

***Structural Variant Calling and Analysis***—We also considered structural variants which joined disparate aspects of the HPV16 genome. To distinguish these form HPV16–human structural variants (i.e., integration breakpoints), we will refer to these events as HPV–HPV structural variants. HPV–HPV structural variants were identified from the svABA analysis as described above. HPV–HPV structural variants were only considered if they passed svABA quality filters. Additional filtering to assess for biologically relevant structural variants was performed based on variant allele frequencies (VAFs) estimated by svABA. The thresholds for VAF filters are described in the results section.

***Viral Deep Copy Loss Analysis***—Considering that HPV+ cancer cells often contain multiple copies of the viral genome, the loss of genomic material in some copies of the virus might result in a decreased copy number (shallow loss) of some viral genes. However, loss of genomic material in all or the vast majority of copies, or deep copy loss, would be expected to result in the most prominent functional differences. In this manuscript, we focused on copy number analysis on regions of deep viral copy losses, which represent the complete or nearly complete loss of the genomic regions in question. To identify these events, nucleotide-specific estimates of read depth were acquired using the Samtools depth function [39]. Relative tumor-specific read depths were calculated as the ratio of read depth to the tumor’s median read depth in the HPV16 regions corresponding to E6 and E7. Nucleotides with read depths <0.2% of median E6/E7 coverage were considered to have deep loss of the involved nucleotides. Nucleotides with <1% of median E6/E7 coverage that bordered on both side areas of <0.2% deep loss were also considered to be a component of the same deep loss event. Deep losses involving only nucleotides in the hypervariable regions 3150–3351 were excluded from analysis as this area had low coverage across the entire data set.

## 3. Results

### 3.1. Integration Analyses

Viral integration sites were identified in approximately 75% of HPV16-associated oropharyngeal (OPSCC, *n* = 104) and uterine cervical (UCSCC, *n* = 44) cancers (see Figure 1A). Individual integration sites, defined as being within a mega-base region of a chromosome, had similar numbers of HPV–human breakpoint junctions per site in OPSCC and UCSCC, with some sites having 10 or more breakpoint junctions (Figure 1B). The viral copy number was not related to the viral integration status of tumors in either OPSCC or UCSCC; however, the viral copy number was higher in OPSCC as compared to UCSCC regardless of integration (Figure 1C). When analysis was restricted to OPSCC and UCSCC tumors with integration, the viral copy number correlated with the number of integration sites (Spearman’s Rho = 0.49, *p* < 5 *×* 10^−^^8^). Amongst tumors with HPV16 integration, OPSCC tumors had significantly more integration sites per tumor (see Figure 1D). Although not statistically significant, integration preferences by chromosome were dissimilar between UCSCC and OPSCC, with chromosome 7 being preferred in UCSCC and chromosome 19 preferred in OPSCC (see Figure 1E). The distribution of sites differed significantly from a random model in OPSCC (Figure 1F), and based on chromosome size, integrations in chromosome 19 were independently enhanced above expectations (chi-squared test, *p* < 0.001). Integrations in areas of the genome with the highest gene density (top 5%) were also enhanced in OPSCC as compared to UCSCC (Figure 1G), and the relative number of genes with integration sites within 500 kB was higher in OPSCC (Figure 1H). Similarly, OPSCC tumors typically had more chromosomes affected by integration sites (see Figure 1I).

Individual sites were mapped to the closest gene, based on the median breakpoint position (see Figure 1J). In agreement with prior reports, recurrent integration sites were noted in both OPSCC and UCSCC. The most common genes associated with integration sites are displayed in Figure 1J or Figure 2. The genomic distribution of individual integration sites is displayed in Figure 2. A gross preference for peri-centromeric integrations were noted (see Figure 2) [43].

### 3.2. Deep Copy Loss Analyses

Deep copy-losses within the HPV genome were identified based on read depth as compared to E6 and E7 in the same tumor (see Methods). The regions of deep loss identified are displayed in Figure 3A. Deep losses were much more common and larger in UCSCC as compared to OPSCC (Figure 3B,C). Only 2 out of 104 (2%) OPSCC tumors had large-scale losses of > 10% of the viral genome, whereas 15 of 44 (34%) of UCSCC had such losses (Figure 3A). Tumors with deep losses of the HPV genome were usually integrated (Figure 3D), with only one UCSCC tumor with deep loss lacking integration. Deep losses involving E1, E2, and E5 were more common in UCSCC as compared to OPSCC (Figure 3E). E2 was the gene most commonly lost in UCSCC and the HPV gene most differentially lost as compared to OPSCC (Figure 3E). More specifically, 27% of UCSCC tumors and 5% of OPSCC tumors harbored deep losses involving E2. As expected, the upstream regulatory region (URR) containing the major early promoters, as well as E6 and E7 genes, were universally spared from deep losses in both diseases (Figure 3E).

### 3.3. HPV-HPV Structural Variant Analyses

We also investigated structural variants (SVs) within HPV16 itself (HPV–HPV SVs). These variants were identified by the svABA pipeline as passing the default quality filter, and were then further filtered by variant allele frequency to look for potentially biologically relevant variants (Figure 4A). There was a non-significant trend towards increasing numbers of SVs in OPSCC as compared to UCSCC (Figure 4B). Interestingly, HPV–HPV SVs of moderate VAF in UCSCC were mostly identified in tumors without integration, and this relationship was distinct from OPSCC (Figure 4C). The reliability of the SV calls and their potential biological relevance was bolstered by relative copy number alterations in HPV that correlated with the identified HPV–HPV SVs. More specifically, all high VAF HPV–HPV SVs had obvious discontinuities in coverage at the identified breakpoints, which also correlated with the orientation of the break-end pair. Two examples of this are shown in Figure 4D–E, where an HPV–HPV SV is the origin of a relative amplification involving L1 in an OPSCC tumor (Case A) and E2 is partially lost in a UCSCC tumor (Case B). Locations of HPV–HPV SV breakpoints appeared to be non-random and grossly favored regions of L2 and the URR, Figure 4F. Examining HPV genomic regions based on their size also demonstrated that breakpoint locations were not random for OPSCC or UCSCC, with UCSCC favoring breakpoints that involved E2 (Figure 4G). Interesting HPV–HPV SVs that caused a break in E2 were more common in UCSCC as compared to OPSCC (Figure 4H). Considering that UCSCC had both an increased percentage of tumors with deep loss of E2 (Figure 3) and a greater number of HPV–HPV SVs causing loss of all or part of E2, UCSCC has a much larger percentage of tumors with E2 alteration as compared to OPSCC (Figure 4H).

Taken together, these results indicate that both integration and HPV–HPV SVs are selected for E2 loss in UCSCC, but not OPSCC. Conversely, the higher frequency of integration sites per tumor and preferential integration in a gene-dense area of the genome, suggests that integration sites in OPSCC are selected for the oncogenic properties resulting from effects on the host genome.

## 4. Discussion

Our direct comparison of HPV integration in oropharyngeal and cervical tumors revealed several important dissimilarities supporting the idea that HPV-driven carcinogenesis is different between these cancers. Below, we summarize major points highlighting their discrete biology.

***The Different Role of E2 in HPV Carcinogenesis***—Integration of the HPV genome into the human genome is an established hallmark of HPV-mediated carcinogenesis. Given that the HPV E2 protein negatively regulates expression of viral oncoproteins E6 and E7, another hallmark of this model is E2 loss. This HPV carcinogenesis model was developed from analysis of UCSCC, and our analyses of UCSCC are supportive since integration was detected in the vast majority and E2 was the HPV gene most commonly altered by deletion (30%) and SVs (40%). On the other hand, integration was less frequent in OPSCC, and even in integrated tumors, E2 loss by deep deletion or HPV–HPV SVs was significantly less frequent in OPSCC, where L2 was found to be the HPV gene most frequently altered by deletion and SVs (Figure 3E or Figure 4G). Although more limited in number of tumors analyzed and without direct comparison to UCSCC, Huang et al. noted a similar lack of E2 loss related to viral integration in squamous cell carcinomas of the penis [44]. The indifference toward the loss of E2 and a higher percentage of non-integrated tumors in OPSCC contradict tenets of most accepted models of HPV carcinogenesis, but is not without precedent in the literature. Our group and others have reported that some HPV+ OPSCC tumors lack integrations, maintain all HPV genes, and express E6 and E7, albeit at lower levels [27]. As a refinement of these findings, here, we report that integration can be detected in 75% of HPV+ OPSCC tumors. This estimate is higher than many prior reports, perhaps reflecting the sensitivity of our assay to detect low frequency variants that could be sub-clonally present in cancer cells. However, even with a very sensitive assay, 25% of tumors lacked HPV integration. Groves et al. demonstrated constitutive expression of HPV oncogenes E6 and E7 without E2 loss [45]. Other groups have suggested that HPV integration, or lack thereof, was not correlated with differential expression of E6 and E7 [46,47]. Furthermore, maintained expression of E2, E4, and E5 have been purported to be critical for an alternate carcinogenic pathway in HPV+ OPSCC, which was associated with a lack of genomic integration [48]. The present work further highlights a distinct relationship of E2 to HPV+ OPSCC as compared to UCSCC, to the extent that a significant role for E2 disruption in HPV+ OPSCC should be questioned.

***A Higher Number of HPV16 Integration Sites Per Tumor in OPSCC***—Considering that E2 loss may be less important in providing selective advantages for cancer and premalignant cells in the oropharynx, other roles for viral integration should be considered, especially since we found HPV integration in 75% of OPSCC cases. Furthermore, we found a strikingly higher number of integration sites per tumor in HPV+ OPSCC as compared to UCSCC, suggesting that maintenance of these integrated HPV copies contributes to cancer development or maintenance. Accompanying higher integration sites per tumor, we also found that viral copy number was higher in OPSCC. Although integration itself was not associated with a higher or lower copy number in either disease, the number of integration sites in an individual tumor was correlated to viral copy number. We note that other groups have associated higher viral copy number in tumors harboring episomal HPV; however, our data do not support this association [49].

The ability of HPV integration to alter expression of nearby human genes (within 500 Kb) has been demonstrated in both OPSCC and UCSCC [7,12] with recurrent integration sites identified near genes implicated in cancer such as *MYC* [7,50], *MYB* [7], *PVT1* [51,52,53], *RAD51B* [50,54], *EMBP1* [55], *CD274/PD-L1* [7,56], and SOX2 [7]. Our results agree with these prior reports confirming integration in or near these genes. Our study identified additional non-recurrent, cancer-associated genes in or near integration sites in HPV+ OPSCC including *TGFBR2*, *FGFR2* [20], *PD-L2* [57], *TRAF3* [26], *BRCA1*, *SPOP* [58], and *BCL-2* [59] Our results bolster prior reports that modulation of the function of genes near HPV integration sites is key in the oncogenesis if OPSCC [7]. As other studies investigating HPV integration in UCSCC have found, we noted HPV16 integration near *NR4A2*, *MYC* [60], *LRP1B* [61,62,63], and *MACROD2* [54]. Unfortunately, additional tissue for expression analysis was not available for the vast majority of the analyzed cases; however, there is a strong precedent in the literature that HPV integration can modulate the expression of nearby cancer genes in both OPSCC [7,50,64] and UCSCC [50,60,65]. Our finding that areas of high gene density are more likely to harbor integration sites in OPSCC as compared to UCSCC, as well as the increased number of integration sites per tumor, are both consistent with the hypothesis that the modulation of human cancer genes may carry increased significance in OPSCC as compared to UCSCC.

HPV+ OPSCC is strikingly more prevalent in men as compared to women [66]. In HPV-negative HNSCC, men typically have a worse prognosis, yet loss of the Y chromosome has been associated with poorer outcomes [67]. Our data identify an interesting integration hotspot in chromosome Y near/at *NLGN4Y* (Figure 1F). Previous work has associated this gene with complex neurocognitive traits such as male homosexuality and disorders including autism [68,69]. A homologous gene on chromosome X, *NLGN4X*, has been shown to have non-identical signaling properties despite 97% sequence homology to *NLGN4Y*, which have been used to explain X-linked phenomenon related to this XY-gene-pair [68]. However, we found only scant reports linking *NLGN4Y* to cancer [70,71]. The most frequent integration hot-spot we identified in OPSCC was near *LINC00486*, on chromosome 2; however, it was notably uncommon for these sites to be called by assembly-based reconstruction, so they should be interpreted with some caution. However, HPV16 integration sites in this area have been confirmed and extensively characterized in UCSCC cell lines [72]. In this study, the integration sites were near an enhancer element that is thought to be related to epithelial differentiation [72], and HPV integration has been demonstrated to be more likely to be found in or near enhancer elements [50].

***Mechanistic Considerations Regarding HPV16 Integration***—The strikingly higher number of integration sites in OPSCC as compared to UCSCC per integrated tumor (Figure 1D, median shift of ~5x), merits consideration from a mechanistic perspective. Integration, like all structural variants, necessitate DNA double-strand breaks as a component of the process [73,74]. HPV+ OPSCC has been noted to be particularly deficient in double-strand break repair [75,76,77,78] and this has been related to the relative radiosensitivity of these tumors [79]. Deficiency of homologous recombination [75] and replication stress [80] have been linked to HPV integration and pathogenicity. Work from our group and others have demonstrated that APOBEC plays a key role in HPV-mediated carcinogenesis in the oropharynx [81], and increased expression and activity of APOBEC3B and APOBEC3A, respectively, have been linked to direct actions of HPV oncoproteins E6 and E7 [82]. APOBEC3s have been associated with genomic instability across multiple cancer types [83]. Maybe not surprisingly, APOBEC3A (A3A) protein expression has been strongly correlated with increased double-strand breaks and HPV viral integration [84]. Others have shown that integration also directly correlates with genomic instability [85]. Whether differences in DNA damage repair and global genomic instability between UCSCC and OPSCC may explain the differential rates of integration is unknown but merits additional focused analysis.

Similarly, one could consider if the molecular process of integration is variable between the two diseases, and indeed, at least three structural classes of integration sites have been identified: single viral genomic insertions (Type I), insertions of tandem viral repeats (Type II), and tandem repeats of hybrid viral–human DNA (Type III) [72]. Unfortunately, without long reads or optical techniques [72,85], it is impossible to distinguish these types, as in our data. However, integrations were commonly found to be quite complex in both OPSCC and UCSCC, with four or more human–viral genomic junctions; these do not fit well into any of the described models in the literature. We note that experimental studies on LINC00486-related integration sites were Type III, and the complex/repetitive nature of the site might explain the failure of assembly-based methods in these cases [72].

Our study was limited in the ability to resolve some complexities of integration events, primarily due to the targeted affinity capture technique, which does not allow unbiased analysis of the copy number of small regions of non-target human genomic DNA, as is possible with whole genome sequencing. More specifically, because of this, we were not able to determine if the integration sites detected were related to chromosomal or extrachromosomal DNA, although other groups using whole genome sequencing have reported detecting human–HPV hybrid extrachromosomal DNA in OPSCC tumors [36].

***Clinical Implications of HPV16 Integration in OPSCC***—We did not find any prognostic relationship to HPV16 viral integration in HPV+ OPSCC in this study. However, this is a frequently discussed association in the literature, with most studies suggesting that integration is a poor prognostic marker [21,86], but conflicting reports exist [27,87]. We notice that other DNA-sequencing-based studies have failed to demonstrate prognostic relationship to integration [7,36]. Interestingly, the strongest evidence for HPV integration being a poor prognostic factor has been based on RNA sequencing [21,86], while conflicting studies as well as studies with no association between HPV integrations and survival were based on DNA technology [7,27,36,87]; therefore, we hypothesize that only transcriptionally active integration sites are prognostic.

## 5. Conclusions

We directly compared 44 cases of HPV16+ UCSCC and 104 cases of HPV16+ OPSCC. E2 loss was common in HPV+ UCSCC, mediated by frequent copy number losses or structural variant breakpoints. Conversely, E2 loss was not selected for in HPV+ OPSCC. Tumors with no evidence of integration represented ~25% of both UCSCC and OPSCC cases. Amongst tumors with genomic integration, OPSCC tumors had many more integration sites per tumor, and were more likely to have integration sites in high gene density areas of the genome. These data highlight clear differences between HPV16-associated UCSCC and OPSCC related to the physical state of the HPV16 genome, and there may be implications for distinct HPV-associated carcinogenesis in the two diseases.

## Figures and Tables

**Figure 1 cancers-14-04488-f001:**
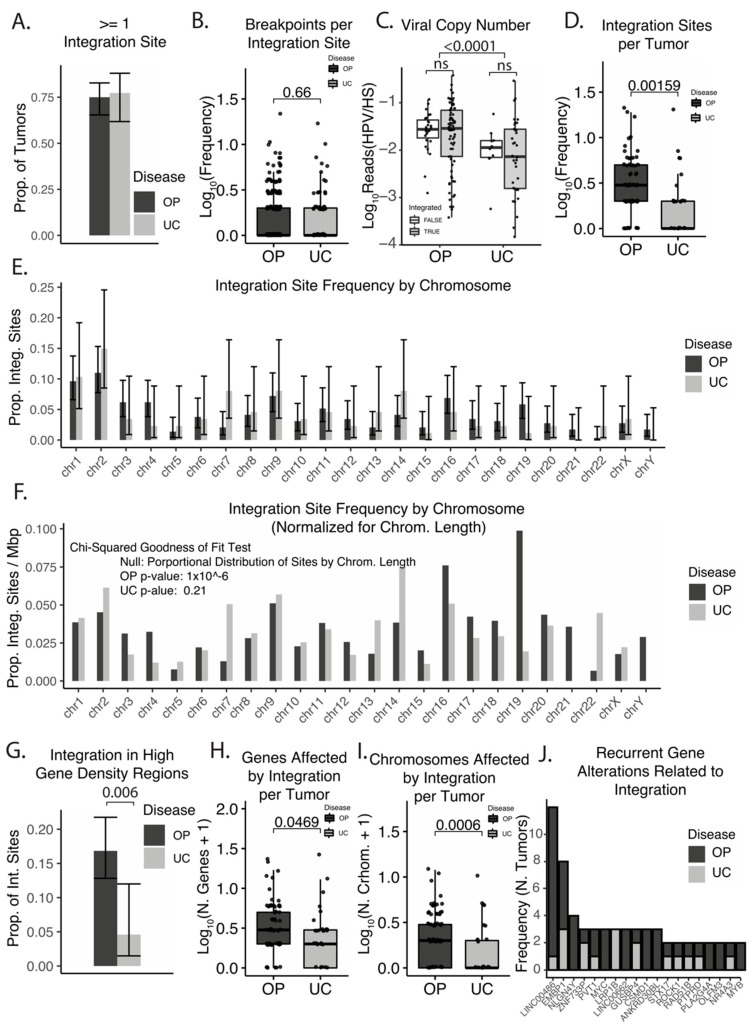
Features of HPV16 genomic integration and integration sites in OPSCC and UCSCC. (**A**) Bar plot—proportion of tumors with detectable integration. Error bars represent 95% confidence intervals. (**B**) Box plot—number of human–viral breakpoints per integration site. Significance based on Wilcoxon Rank-sum test, *p*-values are indicated in plot. (**C**) Box plot—relative viral copy number by disease and integration status. Significance based on Wilcoxon Rank-sum test, *p*-values are indicated in plot. ns—not significant. (**D**) Box plot—number of integration sites per tumor. Significance based on Wilcoxon Rank-sum test, *p*-values are indicated in plot. (**E**) Bar plot—proportion of integration sites by chromosomal location. Error bars represent 95% confidence intervals. (**F**) Bar plot—proportion of integration sites by chromosomal location normalized by chromosome length. Significance based on chi-squared goodness of fit test, with sites’ distribution based on chromosome size as a null hypothesis. (**G**) Bar plot—proportion of integration sites in high gene density regions. High gene density was defined as top 5% of genomic space. Error bars represent 95% confidence intervals. Significance based on chi-squared test, *p*-value indicated in plot. (**H**) Box plot—number of genes affected by integration sites per tumor. Genes within 500 kb of the integration site were included. Significance based on Wilcoxon Rank-sum test, *p*-values are indicated in plot. (**I**) Box plot—number of chromosomes affected by integration sites per tumor. Significance based on Wilcoxon Rank-sum test, *p*-values are indicated in plot. (**J**) Bar plot—frequency of genes effected by integration events. Ordered by decreasing frequency across both OPSCC and UCSCC.

**Figure 2 cancers-14-04488-f002:**
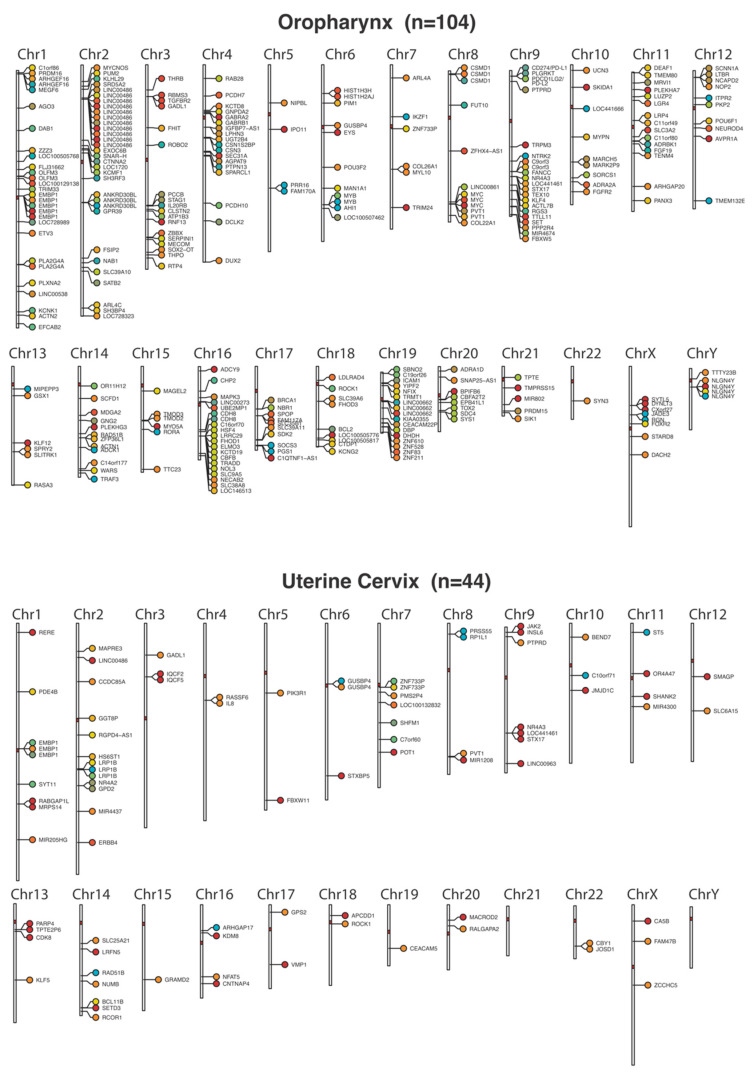
Locations of HPV16 integration sites across the human genome. Genes nearest to the integration sites are indicated. Colors identify tumor samples per chromosome, to allow identification of related sites in single tumors.

**Figure 3 cancers-14-04488-f003:**
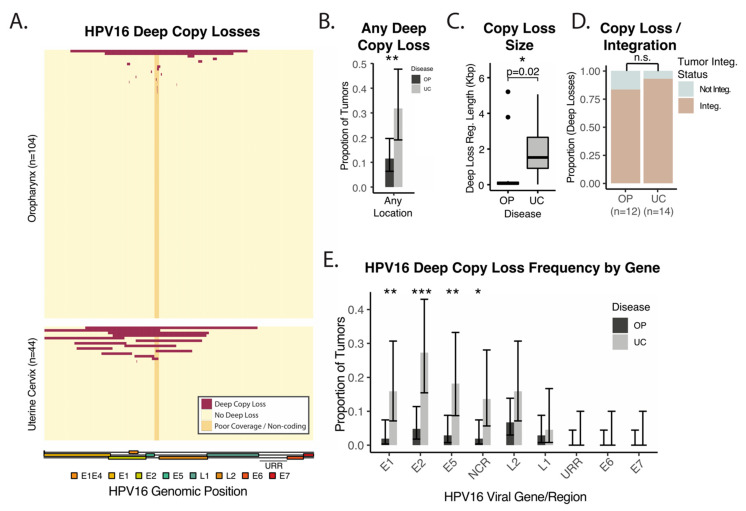
Deep losses of HPV16 genomic material in OPSCC and UCSCC. (**A**) Heatmap—regions of deep loss in the HPV16 genome by tumor. Columns—HPV16 genomic position. Rows—tumors. Dark Red—deep loss. Yellow—no deep loss. Orange—hypervariable non-coding region with poor coverage, excluded from deep loss analysis unless otherwise indicated. (**B**) Bar plot—proportion of tumors with any deep genomic loss in the HPV16 genome. Error bars represent 95% CI. (**C**) Boxplot—size distribution of regions of deep copy loss. *p*-value represents Wilcoxon rank-sum test. (**D**) Stacked bar plot—proportion of tumors with deep loss and viral integration. (**E**) Bar plot—proportion of tumors with deep genomic loss affecting the indicated regions of the HPV16 genome. Error bars represent 95% CI. NCR—hypervariable non-coding region with poor coverage (also orange in Panel **A**). URR—upstream regulatory region. * *p*-value < 0.05. ** *p*-value < 0.005. *** *p*-value < 0.0005.

**Figure 4 cancers-14-04488-f004:**
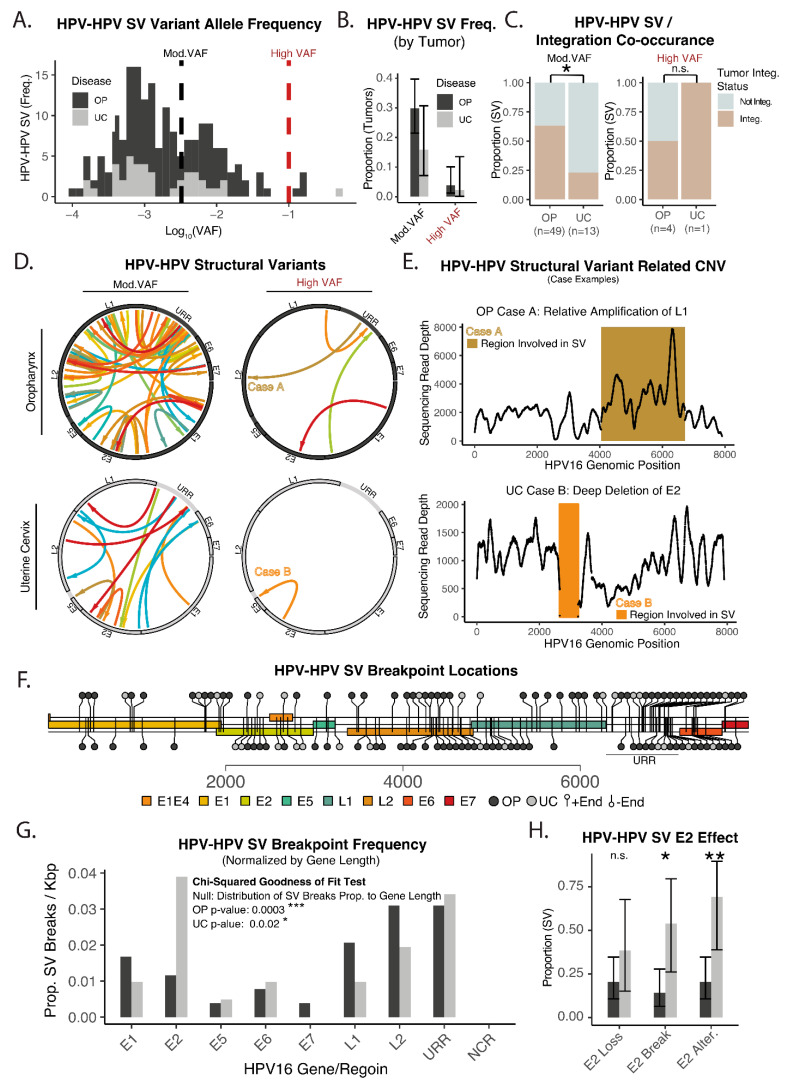
Structural variants in the HPV16 genome in OPSCC and UCSCC. (**A**) Histogram—estimated variant allele frequency (VAF) of HPV–HPV structural variants (SVs). High-quality variant calls were investigated based on empiric VAF thresholds indicated. Black dashed line—moderate VAF threshold; variants with VAF greater than this cutoff were analyzed further. Red dashed line—high VAF threshold, variants with VAF greater than this cutoff were independently analyzed and displayed where indicated. (**B**) Bar plot—proportion of tumors with HPV–HPV SVs. Error bars represent 95% CI. (**C**) Stacked bar plot—proportion of tumors with HPV–HPV SVs and viral integration. Significance testing represents chi-squared test. * *p*-value < 0.05. (**D**) Circos Plots—location and orientation of HPV–HPV SVs. Arrow origin—upstream aspect of structural variant breakpoint junction. Arrow head—downstream aspect of structural variant breakpoint junction. VAF groups are as defined in Panel (**A**). Colors in each individual plot represent a single tumor sample. (**E**) Genomic coverage plots—HPV copy number alterations related to HPV–HPV structural variants in tumors with high VAF SVs. The color and case number link coverage plots to the (left) adjacent circus plots in panel (**D**). (**F**) Lollipop plot—breakpoint locations of HPV–HPV SVs with moderate VAF. Light grey—HPV–HPV SV from UCSCC tumor. Dark grey—HPV–HPV SV from OPSCC tumor. (**G**) Bar plot—proportion of HPV–HPV SV breakpoint sites by HPV16 gene or region. Significance based on chi-squared goodness of fit test, with sites distribution based on gene/region size as null hypothesis. * *p*-value < 0.05. *** *p*-value < 0.0005. (**H**) Bar plot—proportion of HPV–HPV SVs and effects on E2. Error bars represent 95% CI. Significance based on chi-squared test. * *p*-value < 0.05. ** *p*-value < 0.005.

## Data Availability

Raw sequencing data have been deposited to dbGaP; accession no. phs001713.v1.p1.

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
