# Peer review of "Direct Comparison of HPV16 Viral Genomic Integration, Copy Loss, and Structural Variants in Oropharyngeal and Uterine Cervical Cancers Reveal Distinct Relationships to E2 Disruption and Somatic Alteration"

_cancers, 2022, doi:10.3390/cancers14184488_

Round 1

Reviewer 1 Report

Basically, I think this is a good paper, but there are some points that need to be improved. Here are the points which should be improved.

1.                   The introduction needs to be improved. In particular, the difference between the nature of UCSCC and UCSCC should be described more precisely and in a systematic manner.

2.                   In the discussion section, there are many lists of facts and there seems to be no systematic explanation.  Authors should clearly indicate what is the most important conclusion of this manuscript. 

3.                   Considering that it will be read by the general reader, DEEP Copy Loss Analysis should be explained somewhere so that even those who first come into contact with the term can understand it.

Reviewer 2 Report

This  manuscript describes HPV as it is found in oropharyngeal and uterine cervical squamous cell carcinoma: its integration into the genome (or lack thereof), integration sites, copy number, and parts of the HPV genome that are lost.  Although this manuscript treads some of the same ground as recently published papers (references 7, 46, 61, and 85 in this manuscript), it is a valuable independent analysis of HPV integration in OPSCC with a somewhat different focus that includes, importantly, a direct comparison between OPSCC and UCSCC. 

It would be valuable, if possible, to determine whether exosomes -- the focus of ref 85 -- are also present in this study.  Specifically, are the 'integrated' HPV genomes in these tumors stably integrated into the genome, or are they flanked by genomic sequence in exosomes (as described in reference 85)?  Also, why was p16 expression used as an inclusion criterion for OPSCC but not UCSCC?  Please address this potentially conclusion-altering caveat.

Minor comments:

Lines 78-86:  Do references 27-32 really cover the 5 points of difference between OPSCC and UCSCC, including TRAF3 and CYLD mutations?

Lines 136-137:  Were there high-quality reads that couldn't be aligned to the human genome or HPV16? [other viruses, e.g. EBV, or other]

Lines 159-160:  This sentence is missing a verb:  "Integration sites identified from individual HPV-human breakpoints, looking for clusters of breakpoints within 1Mb of each other using in house scripts."

Line 167:  Why are you using "HPV16-HPV16 structural variants" instead of just "HPV16 structural variants"?  If the second "HPV" is necessary, please explain here and in line 275.  If the second "HPV" isn't necessary, please delete it from the section 3.3 title (line 253) and all parts of Figure 4, as well as from the regular text.

 Lines 169-170:  Is "no" in the following sentence a typo for "on", below?

"Additional filtering to assess for biologically relevant structural variants was performed based no variant allele frequencies estimated for svABA"

Also, the sentence above is a good place to define VAF, for example: "based on variant allele frequences (VAF) estimated for svABA."  The first definition of VAF currently comes much too late, in line 281.

Lines 174-175:  Should the second "E6" in the following sentence be "E7"?

"Relative tumor specific read depths were calculated as the ratio of read depth to the tumors median read depth in the HPV16 regions corresponding to E6 and E6"

Line 237:  You should provide, in the text, the loss rate for E2 in UCSCC, for the following sentence to make sense: ”In contrast, less than 5% of OPSCC tumors had deep losses involving E2. "

Figure 3E:  "Proportion" is misspelled on Y axis

Line 269:  "Interesting SVs" – do you mean "Interestingly, SVs . . ."? 

Line 312:  "Our group and others have reported that HPV+ OPSCC tumors that lack integrations, maintain all HPV genes, and express E6 and E7, albeit at lower levels [24]."  As written, this is not a complete sentence.  Removing the first "that" would fix the problem.

Line 338:  "episomal tumors" is unclear; please replace with, for example, "tumors carrying episomal virus"

Line  377:  "Integration, like all structural variants, necessitate DNA double-strand breaks for as a component of the process [70,71]" -- Recommend deleting "for"
